# Prohibiting Babel—A call for professional remote interpreting services in pre-operation anaesthesia information

Gernot Gerger[1], *, Nikolaus Graf[2], Elisabeth Klager[1], Klara Doppler[1,3], Armin Langauer[2], Verena Albrecht[3], Aylin Bilir[2], Harald Willschke[1,2], David M. Baron[2], Maria Kletecka-Pulker[1,3]

1 Ludwig Boltzmann Institute for Digital Health and Patient Safety, Ludwig Boltzmann Gesellschaft, Vienna, Austria, 2 Department of Anesthesia, General Intensive Care and Pain Management, Medical University of Vienna, Vienna, Austria, 3 Institute for Ethics and Law in Medicine (IERM), University of Vienna, Vienna, Austria

☯ These authors contributed equally to this work.

* gernot.gerger@dhps.lbg.ac.at

**Data Availability Statement:** All relevant data are within the paper and its Supporting Information files.

## Abstract

### Introduction

Language barriers within clinical settings pose a threat to patient safety. As a potential impediment to understanding, they hinder the process of obtaining informed consent and uptake of critical medical information. This study investigates the impact of the current use of interpreters, with a particular focus on of engaging laypersons as interpreters, rather than professional interpreters potentially affecting patient safety. A further objective is to explore the reliability of phone-based telemedicine in terms of the retention of important medical facts.

### Methods

In three groups (N per group = 30), we compared how using lay or professional interpreters affected non-German speaking patients' subjectively perceived understanding (understood vs. not understood) and recollection (recollected vs. not recollected) of information about general anaesthesia. Proficient German speaking patients served as the control group. Statistical analyses ($\chi 2$ tests and binomial) were calculated to show differences between and within the groups.

### Results

All three groups indicated similar, high self-reported levels of having understood the medical information provided. This was in stark contrast to the assessed objective recollection data. In the lay interpreter group, recollection of anaesthesia facts was low; only around half of participants recalled specific facts. For patients supported by professional interpreters, their recollection of facts about anaesthesia was significantly enhanced and elevated to the same level of the control group (fluent in German). Moreover, for these patients, providing information by means of phone-based telemedicine before anaesthesia yielded high levels of understanding and recollection of anaesthesia facts.

**Funding:** The author(s) received no specific funding for this work.

**Competing interests:** The authors have declared that no competing interests exist.

## Conclusion

Phone-based telemedicine is a safe and reliable method of communication in the professional interpreter group and German speaking control group, but not in the lay interpreter group. Compared to lay interpreters, professional interpreters significantly improve patients' uptake of critical information about general anaesthesia, thus highlighting the importance of professional interpreters for patient safety and informed consent.

## Introduction

Increasingly multicultural societies and migration in many European countries have led to growing language barriers in healthcare. In the DACH region (Germany, Austria, Switzerland), reports indicate that language barriers affect the routines of healthcare professionals from multiple times a week to daily [1–3]. These barriers can negatively impact informed consent, treatment compliance, healthcare routines, and health outcomes [4–8]. From a legal and ethical perspective, the ability to understand medical consultations is fundamental to self-determined decision-making and informed consent, which is established in various human rights legislation [9–11]. However, in Austria (compared to e.g., Sweden, 12), there is no explicit law granting patients the right to healthcare in their native language, despite the legal responsibility of physicians and hospitals to ensure patient understanding [1,13]. Language barriers are associated with several negative outcomes, including healthcare routines, patient and provider satisfaction, compliance, and access to care and ultimately negatively affect patient safety [14–19]. They can also lead to higher healthcare costs through increased use of tertiary services, multiple visits, and longer hospital stays [5,20–22]. Therefore, effective and reliable interpretation services are crucial for legal, ethical, economic, and especially patient safety reasons.

With respect to interpretation services, the current situation in Austria's healthcare sector is often as follows: The demand for interpretation is undoubtedly high and will likely increase due to demographic developments. This is supported by a recent Austrian study demonstrating that 71% of healthcare professionals interviewed reported encountering language barriers 2 to 3 times a week. In some departments, physicians sometimes treated patients with insufficient command of German to communicate effectively on a daily basis. In only about 37% of these cases are professional interpreters consulted [1]. These rather low levels of professional interpreter use in Austria are in line with international observations [19]. Interestingly, although professional interpreters are often theoretically available at clinics [1], most interpretations are (still) conducted with the help of lay interpreters. These are persons available on an ad hoc basis, either family members such as siblings, (underaged) children or parents, or other non-professional interpreters including friends, non-medical hospital staff or even other patients [1,19,21].

It is interesting to note that public health guidelines often endorse the use of lay interpreters. The Austrian Ministry of Health, for example, explicitly approves the use of laypersons as interpreters to provide pre-operation diagnostic information and obtain informed consent for operations [23]. This fact is critical to note, as using lay interpreters creates challenges in clinical contexts for all parties involved. For example, there are serious implications on the interpreter side: Using non-medical professionals or other available persons (such as other patients or family members) on an ad hoc basis can put severe emotional strain on these interpreters. This factor is considerably aggravated when underaged children function as interpreters

[1,24]. Ultimately, this can increase the caregiver burden and diminish their psychological and physical health [25,26]. Additionally, using lay interpreters raises serious confidentiality concerns [1]. When physicians or other healthcare professionals use their second (or third) language, a further set of problems may often arise: cultural beliefs and traditions of which they are often unaware can compromise the (medical) information provided [27,28]. Non-proficient speakers, in particular, often overestimate their linguistic abilities, assuming they can sufficiently master a language to "get by" without engaging a professional interpreter [29].

The points above raise concerns about the quality of interpretations provided by lay interpreters, and indeed, more negative health outcomes are reported when lay interpreters are used compared to professional interpreters [5,22,29–31]. Lay interpreter services are linked to more interpretation errors [31,32], more emergency room and physician visits [14,33,34], a higher risk of readmission, and an increased length of stay [35]. This gravely impacts patient safety. Thus, although empirical evidence strongly points towards better clinical as well as economic outcomes, the reasons for often not using theoretically available interpretation services deserves further scrutiny. To investigate how the type of interpreter (lay vs. professional) affects patients' understanding and recollection of critical patient safety information, this study considers a typical clinical scenario where informed consent is required: pre-anaesthetic information for general anaesthesia in elective surgery. The information provided about anaesthetic procedures is well suited for testing the effects of language barriers, as it is usually presented in a highly standardized manner following best practice guidelines [36–38]. This allows for a common comparison standard across different operations and patients with different language backgrounds. We developed a questionnaire to evaluate patients' self-reported level of understanding and their knowledge assessment with respect to critical facts about anaesthesia. We hypothesized that the level of understanding and recollection will be lower in the patient group using lay interpreters compared to the group using professional interpreters or a group proficient German speakers (this group included native as well as proficient German speakers).

This study also considered another important aspect: whether telemedical solutions are a reliable basis for establishing informed consent [39]. This is important to demonstrate, as several national, European, and global agendas promote the use of telemedical consultations in clinical contexts [40,41], and the COVID-19 pandemic has clearly shown the need for such solutions [42]. The feasibility of obtaining consent via telephone in anaesthesia settings has been demonstrated in previous studies [43,44], but it needs to be shown that remote consultations result in valid and reliable information transfer that allows patients with and without language barriers to establish informed consent. Low levels of subjectively reported understanding and/or low levels of recollection would indicate problems with this method of providing medical information.

In summary, this study aims to investigate the impact of interpreter type (lay vs. professional) on patients' understanding and recollection of critical medical information, using the context of pre-anaesthetic information. It also examines the reliability of telemedical solutions for establishing informed consent in the presence of language barriers. The findings from this research will provide valuable insights into the importance of professional interpretation services and the feasibility of telemedical consultations in ensuring patient safety and informed decision-making, especially for linguistically diverse patient populations.

## Material and methods

Ethical approval according to the guidelines of the Declaration of Helsinki [45] was provided by the Medical University of Vienna Ethics Committee (2138/2017). The manuscript is drafted

according to the Strengthening the Reporting of Observational Studies in Epidemiology standards (STROBE) [46].

## Participants

The study was undertaken from June, 1st, 2021 to December, 31st, 2021 at the General Hospital of the Medical University of Vienna, Austria, during COVID-19 pandemic restrictions. Pre-anaesthesia information was provided by phone to avoid personal contact and the risk of infection. This was the first use of telemedical services for the provision of pre-anaesthesia information. Participants were patients scheduled for elective surgery from various surgical units (e.g., internal, orthopaedic, etc...) with little or no expected complications (ASA PS < = 2; 47,48) and scheduled for surgery under general anaesthesia. Age was restricted from 18 to 85 years. Only patients were able to give consent on the basis on previous clinical visits and on information available in their medical records were included in the study (e.g. patients with legal guardianships, severe cognitive deficits, etc. were excluded). According to Austrian regulations and internal procedures at the study hospital, anaesthesiologists are responsible for providing and obtaining informed consent for anaesthesia. Surgical information and informed consent for the surgical procedure must be provided separately by the surgical units. This study only looked at the anaesthetic information. The study was conducted until 30 participants were reached in each group. To enable a critical comparison of utilising lay vs. professional interpreters in medical translations, the study included patients who lacked German language proficiency. It is the responsibility of the physician to assess whether informed consent can be obtained from such patients, or whether the language barrier represents an impediment and therefore constitutes a liability risk [13]. Consequently, when checking language skills in the context of telemedical consultations (as in our study setting), it is the duty of the physician to determine whether the patient's language skills are sufficient to understand the consultation to the extent required, as is the case with face-to-face consultations. Interpreters were used for study participants for whom the physician could assume, based on previous medical experience, that informed consent could not be obtained by providing information in the language of treatment. The non-German speaking patients were randomly assigned to two groups (random sequence of professional interpreter and lay interpreter usage) using the randomizer tool (https://www.randomizer.at/). One group was supported by professional interpreters while being given pre-anaesthesia information by the physician. For the other group, laypersons (e.g., family members, friends, etc.; see Sup. B for complete list) served as the interpreters. Both groups were supported by professional interpreters when answering the follow up study questionnaire. All professional interpretation services were provided by SAVD, an agency that provides on-site and remote interpreting services (SAVD, www.savd.at). A third group of proficient German speaking patients (either native German or proficient in German) served as control, thereafter referred to as "proficient speaker".

## Questionnaire

Providing medical information in a way which allows patients to make informed decisions and follow medical instructions requires patients to understand and remember important medical details. The questionnaire was designed to capture these aspects by organising 20 questions into three thematic groups. One set of questions (*Subjective level of understanding*: three questions) surveyed the subjective level of understanding (Do you understand everything? Are you missing information? Do you have any open questions?). These types of questions are typically asked by physicians as a proxy to infer understanding in their patients [4].

The next set of questions tested how well participants had memorised specific facts about general anaesthesia (*Remembering facts about anaesthesia*: Nine questions). Four questions

asked for a single fact (Spontaneous breathing during anaesthesia? Food intake before anaesthesia? Drinking before anaesthesia? Food intake after anaesthesia?) and four questions for multiple facts (Please provide reasons not to eat after anaesthesia; What are the side effects of breathing tubes? What are the side effects of anaesthesia? Which physiological signals are monitored?). Additionally, as a control question, participants indicated whether they had educated themselves about anaesthesia (yes, no), as this could influence recollection of facts about anaesthesia.

The third group included questions designed to evaluate whether physicians had provided specific explanations or asked about certain patient conditions (*Did the physician explain/ ask. . .?)* Seven questions: Type of anaesthesia; possible side effects; current diseases; allergies; previous operations; current medications; chronic pain.

## Procedure

We planned this research to represent a maximally ecologically valid setting. We kept the anaesthesia information procedure in line with standard procedures of the study hospital; that is, the pre-anaesthesia information procedure was hardly affected by the experimental design. Maintaining such a high level of ecological validity can come at the cost of (high) experimental control, however, it comes at the profit of higher generalizability and a better representativity of the status quo of clinical procedures in the study clinic, which was the main focus of this study.

The study was divided into two parts. Firstly, participants received standardised information following pre-anaesthesia guidelines [36] about general anaesthesia (e.g. this included facts about anaesthesia see Results 2, allergy and pre-medication information, see Results 3) from anaesthesiologists (four in total, NG, DMB and two further physicians) via phone. The anaesthesiologists were free to adapt interviews according to the patient's specific surgery, concerns, and information needs. This is in line with the daily pre-anaesthesia information procedure at the study hospital. Graf provided the majority of pre-information (86 participants), followed by Baron (2 participants) and 2 further physicians (2 participants in total). Due to necessities of clinical procedures only physicians associated with the study conducted the interviews. Hence, blinding of physicians to study conditions was not possible.

All participants were called during opening hours (weekdays 7:30–14:30) at the anaesthesia outpatient clinic at the Department of Anaesthesia, Intensive Care Medicine, and Pain Medicine and participants were aware that they will be receiving a call on given day. Where participants (or lay interpreters) were unavailable, a fixed time slot was scheduled.

Regarding assessment of language proficiency and whether an interpreter is necessary this lies in the responsibility and evaluation by the physicians treating the patients (see also Paragraph Participants). As patients undergo several contacts with the medical teams before an elective surgery is scheduled language problems are documented leading to the utilisation of lay or professional interpreters. For this study, depending on this documentation and on random group assignment, a layperson or a professional interpreter was brought in to communicate the pre-anaesthesia medical information in the language of the respective patient. Only participants with languages supported by the professional interpreting service (SAVD, www. savd.at) were included (e.g. BKS, Turkish, Romanian, Polish, Albanian, Arabic, Farsi). In total the pre-information procedure took about in between 15 to 40 minutes with longer durations if an interpreter was needed. Due to a documentation error while conducting the study, regretfully, no exact durations or exact language distributions of the participant sample are available.

Secondly, after physicians had finished providing medical information, the call was transferred to a researcher (VA). The researcher explained the study procedure and asked for informed consent for the study (with the help of professional interpreters when speaking to

non-German speaking patients), documented consent in a separate form, which was followed by a structured questionnaire (as outlined above). As this study was conducted during COVID 19 peak times oral consent (in accordance with the IRB) was the most feasible and safest way to obtain consent. The study continued until 30 participants consented to participate in each condition. Answers to the questions were directly transmitted to an answer sheet by the researcher. After completing the questionnaire, participants were thanked and debriefed. The second part took about 20 minutes.

## Statistical analysis strategy

All statistical analyses were conducted in R and R-Studio [49,50]. All answers were transformed into a binary answer format. For the questions concerning "*Subjective level of understanding*", "*Did the physician explain/ ask. . .?*", and "*Remembering single facts about general anaesthesia*", binary outcomes were given (yes vs. no; recollected vs. not or incorrectly recollected). In the section "*Remembering facts about anaesthesia*", multiple correct answers could be provided for four questions. Initially, it was planned to use the total score of correct/incorrect answers per participant for analysis. However, answering patterns showed that participants recalled between 1 to 4 facts which were all correct, or did not recall any facts at all. Thus, these questions were also transformed into a binary format–participants recalling at least one correct fact (recalled) vs. participants not providing any correct answer (not recalled).

As answers were coded in binary format; proportions, frequencies, percentages, and cross-tables are reported. In testing for significant differences between and within groups, Pearson $\chi2$ tests and binomial tests are reported respectively. We used the ggstatsplot package [51] which allows us to show raw data combined with visualisations of data centrality, dispersion, and directly including outcomes of statistical tests. Results are provided according to the sequence of thematic clusters in the questionnaire: *Subjective level of understanding*–Results 1; *Remembering facts about anaesthesia*–Results 2; *Questions/explanations provided by physicians*–Results 3. Questionnaire data are available in Sup. B.

## Results

### Participants

Thirty participants were included in each group, totalling 90 participants (Age: mean = 49.2, SD = 16.8, Range 18 to 82 years, Sex: 49 female). The three groups did not differ in terms of sex distribution ($\chi2(2) = 1.50$, $p = .47$) or age ($F(2,83) = 0.5$, $p = 0.61$; reduced dfs' in age due to missing values). A priori power calculations using G*Power [52] for the planned $\chi2$-Square tests showed that medium effects ($w > = 0.34$) can be detected assuming a power of 0.8 and an $\alpha$ of .05 ($df = 2$; N = 90).

### Results 1: Subjective level of understanding

Nearly all participants ($> = 97\%$) reported a good understanding of the anaesthesia information provided, had no open questions, and reported not missing any information–See Fig 1.

Following these high levels of understanding, similarly high levels of recall of the provided anaesthesia facts would be expected–see Results 2.

### Results 2: Remembering facts about anaesthesia

Figs 2 and 3 show group comparison results (lay vs. professional interpreter vs. German speaker) and statistical tests for the questions concerning anaesthesia facts.

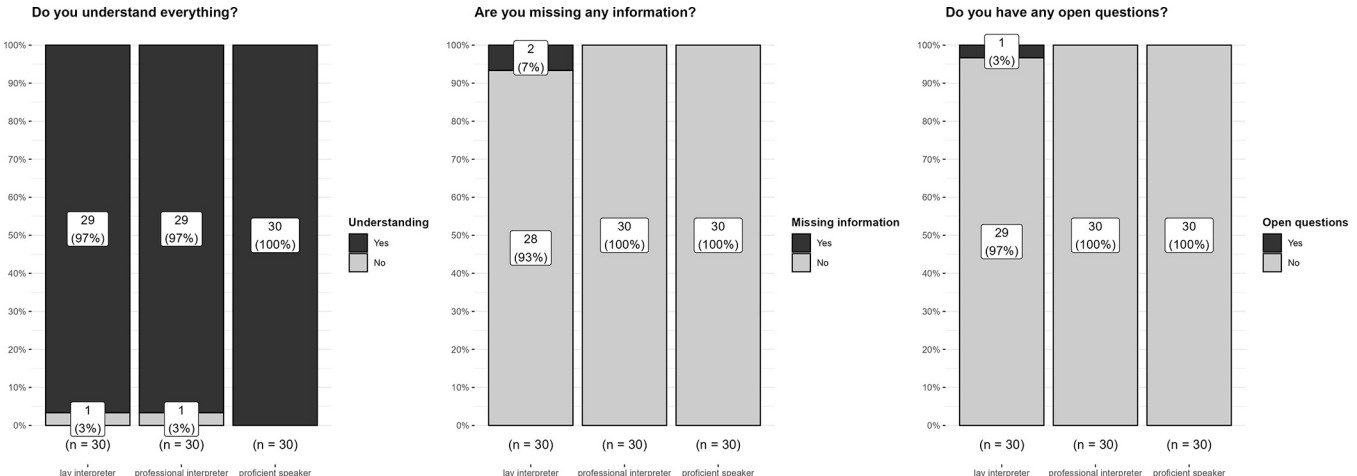

**Fig 1. Subjective level of understanding (3 questions): Bars show absolute (numbers) and relative (%) rates for the binary yes (black)/ no (grey) option.**

As can be seen in Figs 2 and 3, the $\chi 2$ tests show significant group differences. Critically, within the lay interpreter group, the pattern of answers indicated little recall of facts about anaesthesia (rate of recall 27% to maximum 67%). Strikingly, the proficient speaker and professional interpreter group clearly recalled anaesthesia facts well, and here rates of recall were very similar (rate of recall: 73% to 100%). These outcomes provide a clear indication: Patients using lay interpreters had great difficulties recalling important aspects of the general anaesthesia information provided by their physicians. Patients using professional interpreters performed as well as patients who speak German fluently.

In a side analysis, we tested outcomes in participants who had indicated having educated themselves about anaesthesia prior to the consultation (e.g., via internet, books, TV, other individuals, previous surgeries, etc.), as this could influence the level of recollection. Overall, the rate of self-education differed significantly between groups ($\chi 2(2) = 6.02$, $p = .04$, *Cramers-V* = .26). In particular, patients who speak German fluently reported higher rates of self-education compared to the other groups (rate of self-education 53%, *standardized residuals* > 2). Within the lay and professional interpreter groups, levels of self-education were similar (lay interpreter: 23%, professional interpreter: 33%). Consequently, the observed differences in these two groups cannot be attributed to differences in the rate of self-education. This conclusion is supported by the below additional explanatory analyses:

### Exploratory analysis: Can patients self-educating explain differences in recollecting facts about anaesthesia?

In an exploratory approach, we conducted a more detailed analysis by splitting participants along their indicated level of self-education (see Table 1) for questions about single anaesthesia facts. We report descriptive statistics including counts and percentages, and abstained from significance tests, as this was rather exploratory. With respect to the rate of not being able to recollect / incorrectly recollecting anaesthesia facts, two outcomes are worth noting: For participants indicating no self-education, this rate was low in the German speaking and professional interpreter group (not recollected: 0% to 21%). However, at 48% to 70%, this rate was considerably higher in the lay interpreter group. Critically, even if participants in the lay interpreter group indicated having educated themselves about anaesthesia, this rate was still high, falling between 29% to 43%. For the proficient speaking and professional interpreter group,

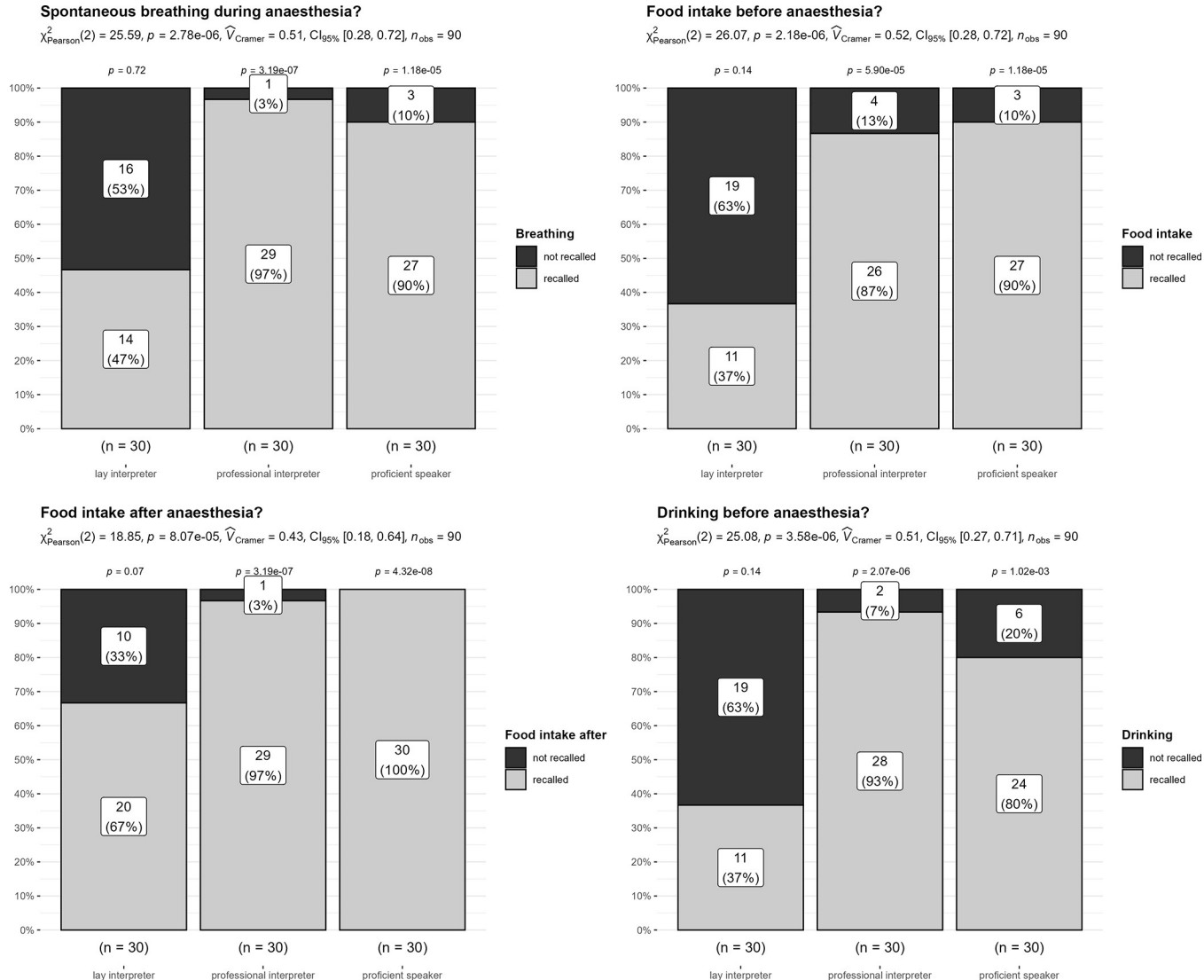

**Fig 2. Remembering facts about anaesthesia: Single fact (4 questions): Bars show absolute (numbers) and relative (%) rates of recollection (recollected: Black; not recollected: Grey).** Additionally, $\chi 2$ test results including effect size measures (*Cramers_V*) and 95% confidence intervals are displayed for the between groups comparison. The p-values above each bar are significance values for binomial tests (50% chance as reference value) within each group.

this rate was again low (not recollected: 0% to 18%). Taken together, this clearly indicates there are additional benefits to using professional interpreters.

## Results 3. Questions/explanations provided by physicians

With respect to the final set of questions (*Which questions/explanations were provided by physicians*?), both the proficient speaking and the professional interpreter groups reported very high rates of received information. Only 3% of the participants in these two groups missed some information–see Table 2. For the lay interpreter group, however, missing information was reported more frequently (23% no explanation about type of anaesthesia; 30% no explanation about possible side effects; 7% no questions about current diseases).

One question resulted in an exception to this general pattern: With regard to chronic pain, all groups reported higher rates of not being asked about this aspect (10% professional

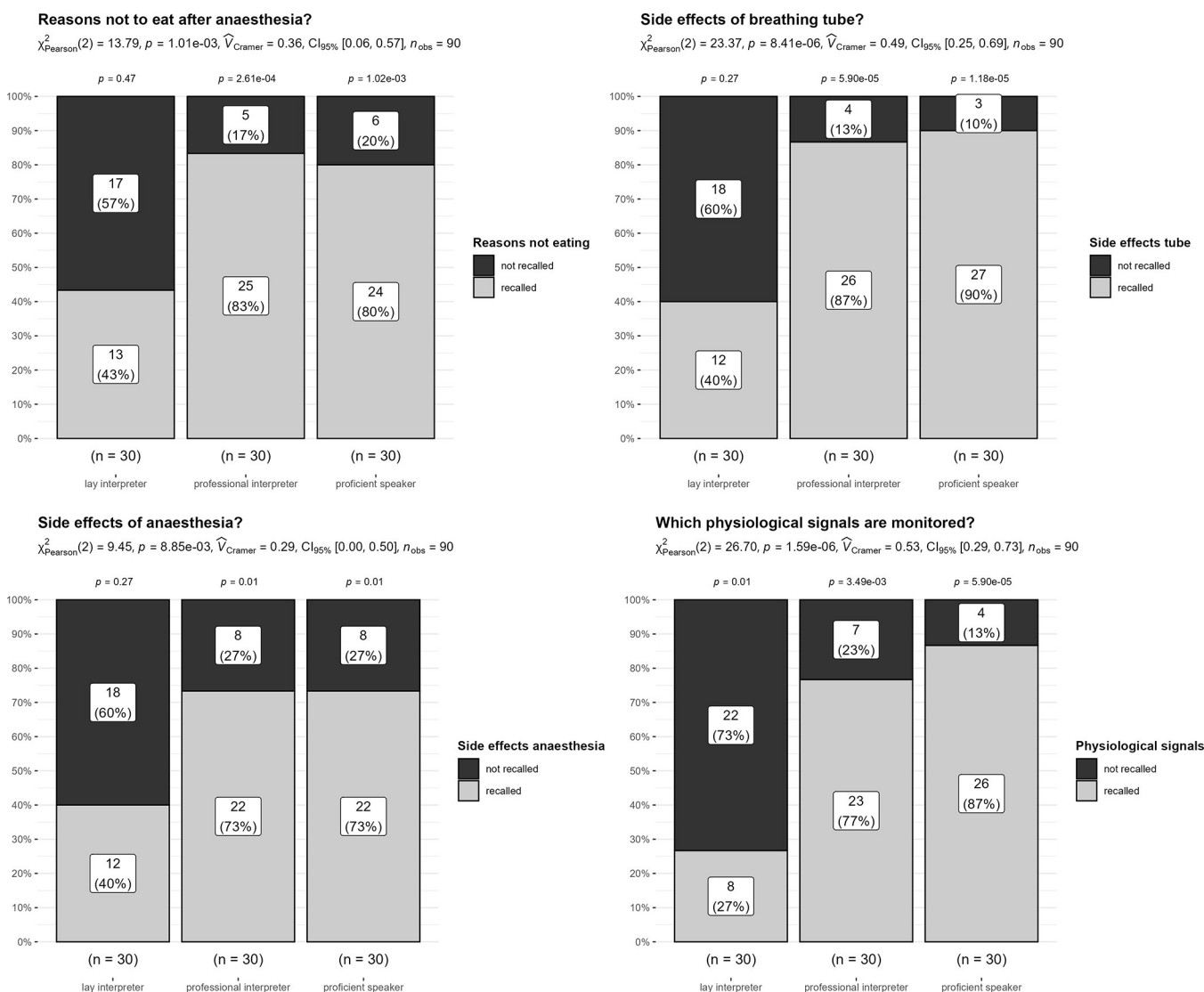

**Fig 3. Remembering facts about anaesthesia: Multiple facts (4 questions): Bars show absolute (numbers) and relative (%) rates of recollection (recollected: Black; not recollected: Grey).** Additionally, $\chi^2$ test results including effect size measures (*Cramers_V*) and 95% confidence intervals are displayed for the between groups comparison. The p-values above each bar are significance values for binomial tests (50% chance as reference value) within each group.

interpreter group, 36% proficient speaking group, and 50% in the lay interpreter group). In sum, while German speaking patients and patients with professional interpreters reported high levels of having received and been asked about clinical information, patients with lay interpreters appeared to have missed some information.

## Discussion

The primary aim of this study was to demonstrate the implications of engaging lay interpreters, currently the predominant method used to address language barriers in hospitals in Austria [1]. The main results are explicit and direct: Patients using lay interpreters were barely able to reproduce critical facts about anaesthesia immediately after having received them (see Results 2). A lot of critical information is literally lost in interpretation. Support from professional interpreters, however, significantly improved patient comprehension and

Table 1. Cross-table: Self-education x recollection for single facts.

| | Self-educated | Eating before? Re-collected | Not re-collected | Drinking before? Re-collected | Not re-collected | Eating after? Re-collected | Not re-collected | Self-breathing? Re-collected | Not re-collected |
|---|---|---|---|---|---|---|---|---|---|
| Lay interpreter | Yes 7 (23%) | 4 (57%) | 3 (43%) | 3 (57%) | 4 (43%) | 5 (71%) | 2 (29%) | 3 (43%) | 4 (57%) |
| Professional interpreter | 10 (33%) | 9 (90%) | 1 (10%) | 10 (100%) | 0 (0%) | 10 (100%) | 0 (0%) | 10(100%) | 0 (0%) |
| Proficient speaker | 16 (53%) | 13 (82%) | 3 (18%) | 13 (82%) | 3 (18%) | 16 (100%) | 0 (0%) | 15 (94%) | 1 (6%) |
| Lay interpreter | No 23 (77%) | 7 (30%) | 16 (70%) | 8 (35%) | 15 (65%) | 15 (65%) | 8 (35%) | 11 (48%) | 12 (52%) |
| Professional interpreter | 20 (67%) | 17 (85%) | 3 (15%) | 18 (90%) | 2 (10%) | 19 (95%) | 1 (5%) | 19 (95%) | 1 (5%) |
| Proficient Speaker | 14 (47%) | 14 (100%) | 0 (0%) | 11 (79%) | 3 (21%) | 14 (100%) | 0 (0%) | 12 (86%) | 2 (14%) |

understanding. In fact, using professional interpreters elevated recollection and understanding of medical anaesthesia information to the same level as in the proficient speaking control group.

In detail, we identified critical differences between the three groups examined. Those utilising lay interpreters encountered substantial challenges in recalling crucial details (such as side effects and safety information). Both proficient speaking patients and patients aided by professional interpreters demonstrated high proficiency in recalling information, and performed equally well. Notably, these objective memory assessments sharply contrasted with the subjective self-reported level of understanding (see Results 1 and 2). In this aspect, nearly all participants from the three groups expressed confidence in having understood the information provided, not having missed any information, nor having any open questions. Such a disparity between subjective perception of knowing vs. objectively knowing is a widely examined psychological phenomenon [53]. Multiple, not mutually exclusive, theories can be proposed to explain this discrepancy, particularly within the lay interpreter group. Factors such as cognitive bias and distortions (e.g., being overconfident or lacking confidence about one's own memory

Table 2. Did the physician explain/ask for . . .?.

| | | Lay interpreter | Professional interpreter | Proficient speaker |
|---|---|---|---|---|
| . . . the type of anaesthesia? | yes | 77% | 97% | 97% |
| | no | 23% | 3% | 3% |
| . . . possible side effects? | yes | 70% | 97% | 97% |
| | no | 30% | 3% | 3% |
| . . . current disease? | yes | 93% | 97% | 100% |
| | no | 7% | 3% | 0% |
| . . . allergies? | yes | 97% | 97% | 100% |
| | no | 3% | 3% | 0% |
| . . . previous operations? | yes | 97% | 100% | 100% |
| | no | 3% | 0% | 0% |
| . . . current medications? | yes | 97% | 100% | 100% |
| | no | 3% | 0% | 0% |
| . . . chronic pain? | yes | 50% | 90% | 64% |
| | no | 50% | 10% | 36% |

abilities), hierarchical and social structures, as well as cultural norms (e.g., reluctance or embarrassment about admitting a lack of understanding, norms within the physician-patient interaction) might all contribute to the observed discrepancy [53–56].

Irrespective of the reasons, the implications of this finding are profound for procedures and workflows in clinical contexts. Frequently, subjective patient self-reports are the sole basis for confirming understanding and comprehension of medical explanations. Our study unequivocally shows that such subjective reports are not a reliable predictor of comprehension and insight. From a patient safety perspective, verbally (or even non-verbally) signalling understanding might not be sufficient proof of the high level of comprehension which is the fundamental basis for informed medical consent. From a legal perspective, in Austria the burden of proof that medical explanations have been understood lies with the physician or care provider [1]. Therefore, in legal terms, the conventional procedure of asking a patient if they have understood might not be adequate as an assumed basis for informed consent. Consequently, when there is doubt about having established understanding during a physician-patient interaction–especially where the risk of incomprehension is high (e.g., due to using lay interpreters)–subjective patient reports should be supplemented by explicitly verifiable questions. Although this approach might require extra time and resources [44,45], only through this procedure can high levels of patient comprehension and understanding be ensured. Ultimately, this contributes to improved patient safety and reduces the clinical, ethical, legal, and economic ramifications associated with language barriers [22,30,31].

Reliably providing information about clinical procedures is a prerequisite for ensuring patient safety. Physicians often use guidelines to achieve this goal (e.g., Thieme-Compliance, & Diomed; AN01E, see 33). However, in daily clinical procedures, physicians sometimes tend to deviate from guidelines as they adopt the information they provide according to medical and patient needs and the current situation. Deviating from guidelines could result in some information being missed. Nonetheless, our study showed that physicians provided information reliably. The groups of proficient speakers and patients with professional interpreters reported that physicians had informed / asked about critical anaesthesia-related conditions well (see Results 3) and showed high recollection levels (Results 2). However, patients supported by lay interpreters more often indicated that they had not been informed about or not asked about certain facts. Whether this was due to constraints in using lay interpreters, general problems of understanding, memory problems in patients, or whether physicians did not or only incompletely explain procedures due to language barriers or other organisational or time constraints (interpretations often afford more time especially with lay interpreters), cannot be directly answered by this study. Patient-physician interaction was not monitored due to confidentiality, privacy, and ethical factors.

The second main finding of this study is that when language is not a barrier, the use of telemedical tools (in our case via telephone) achieves very high recall rates from 73 to the ceiling of 100%. This very high recall rate shows that telephone-based telemedicine is safe and reliable for pre-anaesthesia information. Additionally, providing pre-anaesthesia information via phone comes with other benefits. E.g., at the time of the study, external circumstances made telemedical information provision a necessity as the study was conducted during the height of the COVID-19 pandemic. Personal contacts would have posed significant risks to patients and healthcare professionals and the phone based contact mitigated this risks. This suggests that consulting patients via phone is a suitable method for informing and educating patients about full anaesthesia. Consequently, this telemedical method can serve as basis for informed consent for anaesthesia [40].

## Outlook and limitations

The implementation of telemedical call services revealed additional advantages of remote pre-anaesthesia information provision. The provision of information (at home, via phone) was temporally separate from the time at which informed consent was given (in the hospital, prior to the operation). In standard clinical procedures, information and consent is tightly paired in a very strict clinical schedule. Therefore, patients might sometimes feel pressured to provide consent. Temporal separation allows patients to consider the information and risks beforehand [57]. Indeed, anecdotally, some patients reported greatly valuing this possibility. Another advantage is that travel times for patients and/or physicians are reduced [58,59].

There is one important aspect that was not tested in this study, namely, how providing telemedical consultations impacts trust. Establishing and maintaining trust is a central element in effective patient-physician interactions and positively affects clinical outcomes [60,61]. Future studies should consider this aspect in more detail, especially if the provision of information shifts from a personal to more automated format, e.g., using digital applications for the provision of pre-anaesthesia information [43,57,62]. In fact, a central aim of the European Health Strategy (European E-Health Strategy; 40) is to digitalise the provision of medical information for such procedures. Therefore, such implementations should be backed by strong empirical evidence.

One restriction of our study is the inclusion of only a single hospital: this setting was able to provide the infrastructure needed to conduct the study in a time and resource-efficient manner during COVID. We nonetheless think that, in essence, the outcomes of this study are applicable and transferable to other hospitals and healthcare providers. Recent national and international reports show that other hospitals and providers face similar problems and lack the possibilities and/or do not use professional interpretation services [1,9,14].

Finally, we did not collect more detailed socio-demographic data (e.g., educational level, economic level, etc. . .), mainly due to ethical constraints. Although a more detailed analysis could possibly yield deeper insights, reviews show that such research endeavours are complex. Socio-demographic factors interact in a complex fashion [63]. Given our study design and participant numbers, such granular effects could not have been detected. Nonetheless, we observed moderate to large effects for our main variables (for which the study was statistically well powered), and our study results are cross-validated by other language barrier studies [5,14,32,35,64].

## Conclusions

Compared to lay interpreters, using professional interpreters significantly improves the uptake of critical information about general anaesthesia. It is therefore first and foremost an ethical imperative to act on these outcomes, as improved information transfer is directly linked to better clinical outcomes [1,7,14,16,19,65]. Consequently, it increases patient safety. Importantly, it might also be a financial imperative as studies clearly indicate long-term cost benefits of using professional interpreters in medical care settings [30,65].

We strongly recommend raising awareness, improving guidelines, teaching [66] and, importantly, adopting structures and processes in the healthcare setting to overcome language barriers. Physicians sometimes face conflicting interests in their established healthcare routines. Although they often have some awareness of the negative effects of language barriers, they are required to uphold the workflow within the clinical environment with no or few interruptions and forced to treat patients in a time and cost-efficient manner. It is therefore the responsibility of the healthcare provider to enable and enforce structures and processes that allow clinical workflows to incorporate professional interpretation services [67]. Lastly and

importantly, it is also the responsibility of the government and legislature to act on empirical evidence. For example, recommendations on using lay interpreters within Austria's public guidelines–"Providing pre-operation information for elective surgeries—BQLL PRÄOP" (18, pp. 9–10)"–are clearly contested by the strong empirical evidence of our, and other, studies in the literature [1,7–9,11,14,15,28,29,65].

In a nutshell, use professional interpreters in clinical contexts. Anything else could, and should, be considered as negligent and endangering patient safety.

## Supporting information

**S1 File. List of lay interpreters.**
(DOCX)

**S2 File. Response data questionnaire.**
(XLSX)

## Acknowledgments

We would like to thank Joanna Scudamore-Trezek for proofreading the MS.

## Author Contributions

**Conceptualization:** David M. Baron, Maria Kletecka-Pulker.

**Data curation:** Gernot Gerger, Verena Albrecht.

**Formal analysis:** Gernot Gerger.

**Investigation:** Nikolaus Graf, Armin Langauer, Verena Albrecht, Aylin Bilir.

**Methodology:** David M. Baron, Maria Kletecka-Pulker.

**Visualization:** Gernot Gerger.

**Writing – original draft:** Gernot Gerger.

**Writing – review & editing:** Gernot Gerger, Nikolaus Graf, Elisabeth Klager, Klara Doppler, Armin Langauer, Aylin Bilir, Harald Willschke, David M. Baron, Maria Kletecka-Pulker.

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
