## [Decision Letter · Decision Letter 0]

7 Aug 2024

PONE-D-24-06076Prohibiting Babel - A call for professional remote interpreting services in pre-operation anaesthesia informationPLOS ONE

Dear Dr. Gerger,

Thank you for submitting your manuscript to PLOS ONE. After careful consideration, we feel that it has merit but does not fully meet PLOS ONE’s publication criteria as it currently stands. Therefore, we invite you to submit a revised version of the manuscript that addresses the points raised during the review process.

We look forward to receiving your revised manuscript.

Kind regards,

Stefano Turi

Academic Editor

PLOS ONE

Reviewers' comments:

Reviewer's Responses to Questions

**Comments to the Author**

1. Is the manuscript technically sound, and do the data support the conclusions?

Reviewer #1: Partly

Reviewer #2: Partly

2. Has the statistical analysis been performed appropriately and rigorously? 

Reviewer #1: Yes

Reviewer #2: N/A

3. Have the authors made all data underlying the findings in their manuscript fully available?

Reviewer #1: Yes

Reviewer #2: Yes

4. Is the manuscript presented in an intelligible fashion and written in standard English?

Reviewer #1: Yes

Reviewer #2: Yes

5. Review Comments to the Author

Reviewer #1: Thank you for the opportunity to review this very well-written, interesting manuscript. This study assesses the impact of non-professional (“lay”) language interpreters vs. professional interpreters on patients’ understanding and recall of medical information necessary for anesthesia-related medical consent. Study authors found patients receiving professional interpretation had better comprehension and recall of the medical information, and that provision of this information over the phone was feasible for the patients with professional interpretation and those who did not need interpretation services. While offering compelling informtion, this manuscript could be significantly strengthened by more descriptions of the recruitment process and patient populations.

1. Owing to randomization of patients into interpreter groups, please consider using the Consolidated Standards of Reporting Trials (CONSORT) statement (https://www.consort-spirit.org/) instead of the STROBE standards.

2. Please describe the nature of the pre-anesthesia information provided to patients via phone. Was the information provided in a standardized format? How long were the phone calls, and who spoke with the patients and interpreters? Was there a process to assess patients’ cognition and hearing?

3. Please provide more detail about how patients were recruited for this study. Were the patients being seen from a single surgical specialty or by a single surgeon? Were they all expecting general anesthesia, or were they consented to receive other types of anesthesia?

4. Page 7, Lines 170-175: How did the physician assess patients’ language skills and whether they required interpretation? On what grounds did the physician assume that informed consent could not be obtained in the language of treatment (German)?

5. What languages did the patients in the two interpreter groups speak? Did they differ between the two groups?

6. Please describe the characteristics of patients (e.g., age, sex) in each of the three groups. Readers would expect to see this type of information in a Table 1. Could differences in age explain some of the observed differences in understanding and recall?

7. Current Table 1: Would the authors consider renaming the groups as “Lay interpreter,” “Professional interpreter,” and “No interpreter needed”?

8. The very interesting information in Supplementary Information A could be moved to the main text.

Reviewer #2: In background section, it was difficult to follw the plot. The rationale and importance of conducting this study was blurr, due to too many other topics.

In methods section, I could not find inclusion and exclusion criteria and participant selection process.

In results section, I could not find demografic of participants.

In conclusion section, the authors described "Phone-based telemedicine is a safe and reliable method of communication in the professional interpreter group and German speaking control group". However, this study compares layman and professional but didi not compare phone-based telemedicine and other type of interpreting style (such as face-to-face, video interpreting). So I think they cannot say "Phone-based telemedicine is a safe and reliable method ".

6. PLOS authors have the option to publish the peer review history of their article (what does this mean?). If published, this will include your full peer review and any attached files.

Reviewer #1: **Yes: **Camilla Pimentel, MPH, PhD

Reviewer #2: No

---

## [Author Response · Author response to Decision Letter 0]

24 Sep 2024

Dear Dr. Turi

we would very much like to thank you, and two reviewers (Camilla Pimentel, one anonymous) for drawing our attention to critical points regarding our manuscript. We also thank you inviting us the opportunity to submit a revised version.

In the revision we address open points raised by the two reviewers on a point by point basis.In case we did not follow the suggestions of the reviewers we explain our rationale in detail below. Comments by the Reviewers are highlighted by “REVIEWER NR. ” comments by the authors by “AUTHORS”. Original parts of the MS are enclosed in “ “. Some points of critique were raised by both Reviewers, which we will address together. 

We hope that our revised MS is now suitable for publication in Emotion, and we look forwardto hear back from you. On behalf of the authors, Yours sincerely, 

Gernot Gerger

Detailed answer to Reviews: 

AUTHORS: Both reviewers demanded more information and clarification on participant selection and experimental procedure criteria. In detail they addressed the following aspects:

REVIEWER NR1 and REVIEWER NR2 #############################################

• REVIEWER NR 1, point 3, and REVIEWER NR 2 - Participant selection criteria and recruitment 

• AUTHORS: We added now more information on inclusion/ exclusion criteria. Please see, p .8/9, line 165 ongoing: “Participants were patients scheduled for elective surgery from various surgical units (e.g., internal, orthopaedic, etc …) with little or no expected complications (ASA PS <= 2; 47, 48) and scheduled for surgery under general anaesthesia. Age was restricted from 18 to 85 years. Only patients were able to give consent on the basis on previous clinical visits and on information available in their medical records were included in the study (e.g. patients with legal guardianships, severe cognitive deficits, etc.. were excluded).” 

Moreover, REVIEWER NR 2 additionally demanded clarifications on which surgical professions were included. Aneasthesia and surgical units have to provide separate information and ask for separate informed consents. This study focused on anaesthesia. We added the following sentence: “). According to Austrian regulations and internal procedures at the study hospital, anaesthesiologists are responsible for providing and obtaining informed consent for anaesthesia. Surgical information and informed consent for the surgical procedure must be provided separately by the surgical units. This study only looked at the anaesthetic information. “ Please see p. 7 line 171 to 176).

• REVIEWER NR 1, point 6 and REVIEWER NR 2: Descriptives and socio-demographic data for patients: 

• AUTHORS: Due to privacy, confidentiality, and ethical reasons (ethics vota) only age and sex of participant was documented as socio-demographic variable. This information is provided on p .9, line 245 in the MS. REVIEWER NR1 raised the issue of socio-demograhic variables as potential confounds (e.g. age on memory). We therefore included signficance testing of whether groups differed in proportion of sex (Chi-square) or in age (Anova, three groups), please see p. 11, line 289: “The three groups did not differ in terms of sex distribution (**χ**2(2) = 1.50, p = .47) or age (F(2,83) = 0.5, p = 0.61; reduced dfs’ in age due to missing values).” Hence we are confident that these are no confounding factors for this study. Potential limitations of not assessing more finer grained socio-demographic variables are additionally addressed in the discussion section: Please see p.18, line 461 to 468, 1st paragraph. 

AUTHORS: Below are our responses to the points criticised by only one reviewer: 

REVIEWER NR1 ########################################

REVIEWER NR1, point 1: “Owing to randomization of patients into interpreter groups … consider using CONSORT statement”: 

AUTHORS: The experiment was planned as observational study. It was not planned from the outset as a CONSORT-compliant trial. Consequently, we do not believe that the CONSORT criteria for RCTs, e.g. regarding randomisation (only the non-English speaking group is randomised), pre-registration, recruitment of participants, etc., fully apply to our study. We have therefore decided not to apply the CONSORT statement to this study as suggested by REVIEWER 1. However, based on the CONSORT statement, we have added more detailed study criteria for MS, e.g. regarding the randomisation procedure (p. 7, second paragraph), participant recruitment (p. 6, last paragraph), participant language assessment (p. 7, second paragraph and additionally p. 9, second paragraph), stopping conditions (p. 7, first paragraph). We hope that this additional information contributes to the clarity of the experimental procedures and fulfils the REVIEWERS' NR1 request.

REVIEWER NR 1, point 2: More information about the delivery of pre-anaesthesia information: 

AUTHORS: This study was designed to be as ecologically valid as possible, as described on page 8. That is, doctors followed their standard information procedures. The only experimental variation was the random assignment to the lay or professional or professional interpreter group. This procedure ensures that the study is representative of how anaesthesia information is provided in everyday clinical practice at the study hospital. Importantly, as requested by REVIEWER 1, we have now added more information and explanation regarding the justification for high ecological validity (p. 8, first paragraph); standardisation of pre-anaesthesia information (p. 8, last paragraph); (non-)blinding of physicians (p. 9, first paragraph); listing of study physicians (p. 8, last paragraph); assessment of language proficiency (p. 7, second paragraph and additionally p. 9, second paragraph); languages translated (p. 9, second paragraph); and duration of pre-anaesthesia (p. 9, second paragraph). Unfortunately, due to a documentation error, we did not have the numbers for duration and translated language for each participant. Cognition and hearing were not directly assessed during the interviews. The responsibility for assessing a patient's ability to follow clinical information rests with the treating physician. If cognition (or hearing) is impaired, or if a patient is under (legal) guardianship, this is documented in the records and the clinician can act on this information. For this study, only patients who were judged by the doctor to be able to give consent were included. This again followed the standard procedures of the study clinic.

REVIEWER NR 1; point 4 and 5: This regarded how language skills were assessed (REVIEWER NR 1, point 4 and point 5):

AUTHORS: We provided this information on p. 7, 2nd paragraph in the Participants section in the first version of the MS. We now additionally repeat this information to p.9 2nd paragraph, Procedure, to achieve more clarity. Essentially, as outlined above, we have followed standard clinical practice where it is the physician's responsibility to determine whether or not translation services are required.

REVIEWER NR1, point 7: Renaming of categories to Lay interpreter, professional interpreter, No interpreter needed: 

AUTHORS: As suggested, we have changed the non-German categories to lay and professional interpreters. For the German speaking group we now use proficient speaker as this somehow includes both native German speakers and fluent German speakers. We did not use the category 'no interpreter needed', as we felt this would mean that only non-native speakers of German would have been included in the study.

REVIEWER NR1: Move supplementary Information A to main MS: 

AUTHORS: We moved this part as suggested, see p. 12., 3rd paragraph. 

REVIEWER NR2 ###############################

REVIEWER NR2: It is difficult follow the Figures: 

AUTHORS: We added now Figure captions that were missing in the first version of the MS. We hope this clarifies matters. 

REVIEWER NR2: “The rationale and importance of conducting this study was blurr ….”: 

AUTHORS: It seems that reviewer 2 had some difficulty in following the development of the rationale of the study. Interestingly, this criticism somehow contrasts with reviewer NR1's comment that the MS is "... very well written, interesting manuscript". If parts of the MS need clarification, we will address this issue. However, as no details have been provided as to exactly what aspects of the study were not clear to REVIEWER NR2, it is difficult for us to respond to this specific point of criticism. Perhaps one of the problematic parts was the first paragraph of the abstract, where our description of the study aims was rather vague. We have modified the following sentences (italicised parts modified): “ As a potential impediment to understanding, they hinder the process of obtaining informed consent and uptake of critical medical information. This study investigates the impact of the current use of interpreters, with a particular focus on the efficacy of engaging laypersons as interpreters, rather than professional interpreters potentially affecting patient safety. A further objective is to explore the reliability of phone-based telemedicine in terms of retention of important medical facts.” p.2 line 19 to 25, Introduction” 

REVIEWER NR2: Reviewer NR 2 criticized the sentence “Phone-based telemedicine is a safe and reliable method of communication in the professional interpreter group and German speaking control group ". as an overarching claim, as we did not compare phone based telemedicine to other forms of interpreting (e.g. face to face, video). We see that our previous description can be interpreted in this way. However, this was not our intention - we based this claim on the recall of anaesthesia facts, which was 80 to 100%, which is a ceiling effect. This shows that the information is reliably remembered in the expert (German) and professional interpreter groups. We have changed the following sentences: “The second main finding of this study is that when language is not a barrier, the use of telemedical tools (in our case via telephone) achieves very high recall rates from 80 to the ceiling of 100%. This very high recall rate shows that telephone-based telemedicine is safe and reliable for pre-anaesthesia information.p. 16, last paragraph”

---

## [Decision Letter · Decision Letter 1]

20 Oct 2024

PONE-D-24-06076R1Prohibiting Babel - A call for professional remote interpreting services in pre-operation anaesthesia informationPLOS ONE

Dear Dr. Gerger,

Thank you for submitting your manuscript to PLOS ONE. After careful consideration, we feel that it has merit but does not fully meet PLOS ONE’s publication criteria as it currently stands. Therefore, we invite you to submit a revised version of the manuscript that addresses the points raised during the review process.

We look forward to receiving your revised manuscript.

Kind regards,

Stefano Turi

Academic Editor

PLOS ONE

**Additional Editor Comments:**

I would like to thank the Authors for reviewing the manuscript according to reviewers' comments. 

All the comments by reviewer 1 were satisfied.

I would only ask to the Authors to shorten introduction, focusing on the main elements of proposed topic. In my opinion, this could increase the quality of the manuscript, thank you. 

Reviewers' comments:

Reviewer's Responses to Questions

**Comments to the Author**

1. If the authors have adequately addressed your comments raised in a previous round of review and you feel that this manuscript is now acceptable for publication, you may indicate that here to bypass the “Comments to the Author” section, enter your conflict of interest statement in the “Confidential to Editor” section, and submit your "Accept" recommendation.

Reviewer #1: All comments have been addressed

2. Is the manuscript technically sound, and do the data support the conclusions?

Reviewer #1: (No Response)

3. Has the statistical analysis been performed appropriately and rigorously? 

Reviewer #1: (No Response)

4. Have the authors made all data underlying the findings in their manuscript fully available?

Reviewer #1: (No Response)

5. Is the manuscript presented in an intelligible fashion and written in standard English?

Reviewer #1: (No Response)

6. Review Comments to the Author

Reviewer #1: (No Response)

7. PLOS authors have the option to publish the peer review history of their article (what does this mean?). If published, this will include your full peer review and any attached files.

Reviewer #1: **Yes: **Camilla Pimentel, MPH, PhD

---

## [Author Response · Author response to Decision Letter 1]

6 Nov 2024

Dear Dr. Turi

We would like to thank you very much for giving us the opportunity to submit a second revision requesting minor changes. As the points of reviewer 1 were fully satisfied after our first round of revisions, we focused on your suggestion to provide a more concise and shorter introduction to the reader. We have shortened and condensed the introduction by about 30% to 1000 words, 6 paragraphs (previously 1400 words, 9 paragraphs) and hope this is to your satisfaction. 

By the way (according to the PLOS ONE preprint option) this paper has been uploaded to a pre-print server (https://www.medrxiv.org/content/10.1101/2024.02.16.24302964v1). As this is the first time I have uploaded a paper to a preprint server, could you explain how to proceed with the preprint in order to meet the requirements of PLOS ONE? 

We hope that our revised MS is now suitable for publication in PLOS ONE and look forward to hearing from you. 

Yours sincerely, 

Gernot Gerger

---

## [Editor Report · Decision Letter 2]

25 Nov 2024

Prohibiting Babel - A call for professional remote interpreting services in pre-operation anaesthesia information

PONE-D-24-06076R2

Dear Dr. Gerger,

We’re pleased to inform you that your manuscript has been judged scientifically suitable for publication and will be formally accepted for publication once it meets all outstanding technical requirements.

Kind regards,

Stefano Turi

Academic Editor

PLOS ONE

---

## [Editor Report · Acceptance letter]

8 Jan 2025

PONE-D-24-06076R2 

PLOS ONE

Dear Dr. Gerger, 

I'm pleased to inform you that your manuscript has been deemed suitable for publication in PLOS ONE. Congratulations! Your manuscript is now being handed over to our production team.

Kind regards, 

on behalf of

Dr. Stefano Turi 

Academic Editor

PLOS ONE